# Patients, Social Workers, and Pharmacists’ Perceptions of Barriers to Providing HIV Care in Community Pharmacies in the United States

**DOI:** 10.3390/pharmacy9040178

**Published:** 2021-11-02

**Authors:** Adati Tarfa, Kristen Pecanac, Olayinka Shiyanbola

**Affiliations:** 1Division of Social and Administrative Sciences, School of Pharmacy, University of Wisconsin, Madison, WI 53705, USA; olayinka.shiyanbola@wisc.edu; 2School of Nursing, University of Wisconsin, Madison, WI 53705, USA; lund2@wisc.edu

**Keywords:** PLWH, community pharmacy, community pharmacists, HIV, retention, barriers to care, social workers, ADAP, HIV care

## Abstract

Retaining people living with HIV (PLWH) in clinical care is a global priority to end the HIV epidemic. Community pharmacies in the United States have structural influences on the success or failure of retention in HIV care by supporting patients’ complex needs. However, to date, barriers to retention in care in the community pharmacy setting have not been examined beyond pharmacy services of medication therapy management. We utilized the patient-centered medical home model to examine the barriers to HIV care in the community pharmacy setting. We utilized semi-structured interviews to collect data from 15 participants: five PLWH, five community pharmacists, and five social workers from a midwestern state. Interview data were transcribed and analyzed using directed content analysis. Four key themes emerged regarding the barriers that impact utilization of community pharmacy services by PLWH: the perception of the role of community pharmacists in HIV care, perceptions of pharmacists’ HIV knowledge, perceptions of pharmacy operation and services, and negative experiences within the community pharmacy space. Participants’ perceptions of solutions for improving HIV care in the community pharmacy focused on improving the relationship between pharmacists and patients, ensuring that the community pharmacy is a private and safe space for patients, and having a diverse pharmacy staff that is equipped to take care of the diverse and marginalized HIV population, such as transgender people.

## 1. Introduction

### 1.1. Underutilization of HIV Care Services and the Implications of Poor Retention in Care

In the United States, people living with the human immunodeficiency virus (HIV) do not consistently utilize care [1,2,3], resulting in poor health outcomes including death [4]. People living with HIV move along a continuum of care [5,6,7] from not knowing their status to becoming retained in HIV and then virally suppressed [8,9]. The care people with HIV receive is examined on a continuum, where a person moves from not knowing their HIV status to becoming virally suppressed [5,6,7]. Retention in HIV care occurs when patients utilize HIV care services to move along the HIV continuum of care [8,9]. More than 80% of the individuals who are retained in HIV care achieve viral suppression [10]. People living with HIV who are not retained in care experience higher rates of mortality, disability, and antiviral resistance due to missed opportunities of comorbid disease management [11,12]. Comorbid conditions, such as substance use disorder, psychiatric diseases, hepatitis C, diabetes, and heart disease, are detectable and require active management [13,14]. Active HIV management can only be attained when a patient is retained in care. Additionally, when patients utilize HIV care they will benefit from their antiretroviral therapy and receive prophylaxis for opportunistic infections [15,16]. Poor retention in care is a public health threat because 61% of new HIV infections are caused by people who are living with HIV but who are not retained in care [7]. However, despite the benefits of retention in care, only 57.9% of people living with HIV are retained in care [7,17]. To improve retention in care, it is imperative to understand what barriers exist in HIV care settings that make it difficult for patients to remain in care. 

### 1.2. Community Pharmacies as a Readily Available Resource for People Living with HIV

Prior studies have established that the community pharmacy setting in the United States provides critical services for patients’ retention in care by promoting their health and well-being [18]. According to the United States Department of Health and Human Services, pharmacies are critical to success or failure of retention in care by supporting patients’ needs [15]. A community pharmacist is a pharmacist practicing in a registered premise that is approved and licensed to dispense medication, provide professional counseling, and perform pharmacy services to people in a local area [19]. Community pharmacists are not usually specialized in HIV care [20]; nonetheless, they play an important role in HIV care because of their placement in the community and accessibility to patients [20,21]. People living with HIV may have multiple health care providers, but they often receive all their medicines from one pharmacy, making the pharmacist a key point of contact [7]. The pharmacy setting is also considered less stigmatized and an ideal location for HIV care [22,23,24]. Currently, more people living with HIV are receiving care in the community setting rather than the hospital setting [21]. Conveniently, approximately 70% of people in rural areas live within 15 miles of a pharmacy, and 90% of people in urban areas live within two miles of a pharmacy in the United States [22,25]. This placement of pharmacies makes them a readily available resource for people who are less likely to interact with the health care system regularly, preventing patient loss to care [21].

### 1.3. Exploring Barriers to Retention in Care in the Community Pharmacy Environment

People living with HIV depend on the continuous utilization of HIV care services [26]. Prior studies about retention in care in community pharmacies are limited in scope as they have focused solely on medication adherence [21,25,27,28,29,30,31]. The existing gap in our knowledge about community pharmacy-specific barriers to HIV care needs to be addressed. The abundance of community pharmacies along with a critical need to improve retention in care present a rich opportunity to examine retention in care in the community pharmacy setting. The goal of this study was to understand the barriers that impact patient utilization of community pharmacies for HIV care services in the United States. Gaining a better understanding of the barriers and facilitators to retention in care in the pharmacy setting may assist in designing interventions to improve retention in HIV care. To our knowledge, this is the first study in the United States to examine barriers to retention to HIV care in the community pharmacy setting and to utilize three stakeholder groups: patients, pharmacists, and social workers. In addition to recognizing the barriers, the stakeholders in this study provide some suggested solutions on how to address the barriers to HIV care in the community pharmacy in the United States. This paper is part of a larger study that explores the barriers to linkage and retention in care that patients experience within their communities. The aim of the overall study was to identify these barriers and explore ways in which pharmacists and social workers can collaborate interprofessionally to address these barriers. This paper focuses on the barriers to care experienced specifically in the community pharmacy and the recommendations on how community pharmacists can address those barriers.

## 2. Methods

### 2.1. Study Design and Sample

The study used a directed content analysis approach [32] to understand the barriers to retention in care in the community pharmacy setting. Content analysis is a qualitative research technique that enables researchers to interpret meaning from a body of text. Directed content analysis starts with a framework that directs the analysis. In this study, research questions were guided by the patient-centered medical home model (PCMH) [33,34,35], a community-based care system model that aims to provide comprehensive and accessible patient-centered care for persons with HIV [36,37,38,39,40].

This study included participants representing a range of stakeholders: people living with HIV, community pharmacists, and social workers [41] sampled from a midwestern state in the United States. People living with HIV were eligible to participate if they were older than 18 years old, self-reported as living with HIV, and receiving HIV care from a community pharmacy. Using purposive sampling, people with HIV were recruited through word-of-mouth by their infectious disease physicians, pharmacists, nurses, HIV community leaders, and social workers.

Pharmacists were included in the study if they practiced in a community pharmacy setting and had patients that were living with HIV. The researchers sampled pharmacists who were practicing in the community setting and approached them at their place of practice with information about the study. Social workers were included in the study if they had an active caseload of people living with HIV. Social workers were recruited from a state directory of social workers trained as specialists to care for people living with HIV. The social workers were contacted via an email that introduced the study opportunity and information on how to participate. The research team included one community pharmacist (AT) and two researchers with expertise in qualitative research (KP, OS). Two of the researchers are in the field of health services research in pharmacy (AT, OS).

### 2.2. Data Collection

To best understand participants’ perception of barriers and facilitators to HIV care within the community pharmacy setting, face-to-face semi-structured interviews were conducted for each participant from May 2018–August 2018. Interviews were conducted by a trained researcher who took notes as well as recorded audio from the interviews. Each interview took place either in-person at a private location or over the phone and lasted approximately one hour. To generate rich data, probing techniques were used to encourage participants to share more detailed information relevant to the study. The interviews ranged from 40 min to one-and-a-half hours. Only unconcealed sociodemographic participant information—race and gender—were collected to minimize the collection of sensitive information (such as education level and income) that is not pertinent to the study.

Using a directed content analysis approach, when data is collected through interviews, open-ended questions and targeted questions about pre-determined constructs are used [31]. As such, the semi-structured interview guide was designed using adapted constructs of the PCMH model—experiences, activities, interventions, and outcomes—that have an impact on the utilization of HIV care services in the community pharmacy setting. This paper focuses on the experiences and activities constructs to examine barriers to care in the community pharmacy setting. Questions explored patient experiences in seeking care in the community pharmacy setting and the barriers experienced within the environment. Pharmacists shared their experiences providing HIV care in the community pharmacy setting. Social workers also shared their experiences in working with community pharmacists to provide care to people living with HIV and the barriers their patients experience when receiving care within the community pharmacy environment. Activities pharmacists can provide in the pharmacy setting to improve the utilization of HIV care from the purview of the study stakeholders were explored in-depth. In this study, we defined barriers to retention in care as any factor identified by study participants that hindered patients from continuously utilizing pharmacy services and continuously seeking HIV care.

### 2.3. Data Analysis

Directed content analysis [31,42] was used to analyze the data deductively. Directed content analysis starts with a model, existing theory, or research findings to guide data collection and analysis [31,42]. The researchers followed the following steps in their directed content analysis: they started by reading the transcripts several times to achieve immersion; then, they read the data line by line to capture ideas; and finally, they coded and organized the themes and categories in accordance with the mapping of the constructs guided by the PCMH model. Factors that acted as barriers to retention in care were coded into four constructs of the PCMH: (a) interventions (b) activities, (c) experiences, and (d) outcomes. Specific themes emerging under each construct emerged from the data. The identified themes within each construct were explained with direct verbatim quotes to further clarify the themes. The researchers allowed for the development of themes inductively to expand their exploration of the data. The findings presented in this paper focus on the experiences and activities constructs as well as the inductive findings.

The researchers used NVIVO Version 12 (QSR International-Melbourne) to code and manage the transcripts. The notes taken during the interviews were used during the data analysis process to provide additional context to the transcripts and participants’ quotes. Towards the completion of coding the 15 transcripts, the researchers did not see any new themes, the data was saturated, and no further interviews and analysis were needed. Two qualitative researchers (A.T and K.P) independently coded the transcripts and then met together to discuss each code and interpretation to ensure the findings were grounded in the contents of the transcripts. When the researchers did not agree on a code, they consulted a qualitative methodologist, O.S., to review the codes. Codes were iteratively reviewed and further refined by the research team.

#### Ethics and Compliance Statement

This study was approved by the Institutional Review Boards (IRB) of the University of Wisconsin-Madison. The interviewer obtained and documented verbal consent from all participants as approved by the above IRB for this research.

## 3. Results

When exploring the PCMH model construct, including experiences of patients, community pharmacists, and social workers, five themes regarding the barriers that impact utilization of community pharmacy services by people living with HIV emerged: the perception of the role of community pharmacists in HIV care, perceptions of pharmacists’ HIV knowledge, perceptions of pharmacy operation and services, negative experiences within the community pharmacy space, and trainings for pharmacists to consider (Table 1). Under the PCMH model activities construct, participants shared recommendations for change concerning pharmacy staff and the delivery of pharmacy services to improve the utilization of community pharmacy for HIV care. In each section below, we elaborate upon these findings and consider their implications for community pharmacies. We labeled quotes from community pharmacists as CP, social workers as SW, and people living with HIV as PLWH.

### 3.1. Perceptions of the Role of Community Pharmacists in HIV Care

Participants reported diverse perceptions of the role of pharmacists in HIV care based on their experiences when receiving care from a pharmacist, providing care as a pharmacist, or working together with a pharmacist as social workers. Pharmacists were viewed as the drug experts only. The limited perceived scope of pharmacists acted as a barrier to them being patient advocates. Participants in the study including pharmacists view patient advocacy as something outside of their limited scope.

“I don’t think the rest like of care teams understand how important the role of pharmacist is or the pharmacy is, because they are your every month point of contact.” [SW 2].

#### 3.1.1. Pharmacists have a Limited Role in HIV Care as Drug Experts

The study participants praised pharmacists as the most knowledgeable members about drugs on a patient’s care team. Social workers especially stated that they trusted pharmacists to know more about drugs and reach out to them concerning patients’ drug concerns. Social workers saw this role as limiting the scope of the pharmacist’s practice.

“When you think about pharmacists, you think about, you know, like pure medication.”[SW3].

#### 3.1.2. Pharmacists Act as Patient Advocates

Pharmacists shared their experiences in providing care for patients and services they considered to be patient advocacy. They described these roles as part of their practice, however, more associated with surrogate caregivers, such as social workers.

“[…]the pharmacist can kind of almost step in and either allay some of those fears and dispel some of the myths, as well as maybe just kind of step in and be, you know, like a tiny surrogate caregiver.”[CP 3].

### 3.2. Perceptions of Pharmacists’ HIV Knowledge about HIV Resources

Study participants shared their concerns about the varying experiences and knowledge of pharmacists about HIV care. Participants were concerned about pharmacists’ exposure to people living with HIV and their proficiency in addressing cost and cognitive issues. Patients saw that being part of the HIV community improves a pharmacist’s knowledge about HIV.

“(Pharmacists) may be not as confident in their knowledge and background as, say, (name of pharmacist), who had friends who had had AIDS and, you know, was very plugged into the community, so I found that very helpful.”[PLWH 1].

People living with HIV have barriers to care and needs that pharmacists may not be readily aware of. Participants in this study expressed a lack of confidence in the level of knowledge pharmacists may have about patient needs.

“So, right, like I have a fair number of people with cognitive delay. Sometimes people with cognitive delay or with learning disabilities or people with, you know, this is a whole other thing unto itself, but limited literacy, are pharmacists aware of that, and do they ask that question, and how do they perceive, if the person says that?”[SW 3].

Social workers were not confident in the level of knowledge pharmacists have about drug assistance programs for people living with HIV. Pharmacists also acknowledge that some community pharmacists that are not used to dispensing HIV medication may not have the knowledge of prescription assistance programs to cut patient medication costs.

“So it felt like, and I don’t know if the newer generation of pharmacists are doing quite it’s like, oh, your insurance doesn’t work, sorry. You know, it’s like, oh, your insurance doesn’t work, let’s find out why your insurance doesn’t work, you know. So I did a lot of that. And, oh, you don’t think you can afford your meds.”[CP 1].

Pharmacists in this study saw a need to train community pharmacists in insurance processes, especially in pharmacies that do not traditionally serve people living with HIV.

“So sometimes that takes a little bit of teaching the pharmacy that they’re connecting with to, you know, educate them about, okay, what does it mean when somebody has coverage through the ADAP program and how, you know, this is how you run into barriers filling it if this, and coming up with like a huge bill, and then there’s something wrong with the billing[...]”[CP 5].

### 3.3. Perception of Pharmacy Operation and Services

Although pharmacists have accessibility to people living with HIV, there are barriers to the provision of pharmacy services, such as issues with time and staffing. Pharmacists do not have enough time to attend to direct patient care and their administrative roles. Staffing issues also occur in the pharmacy, whereby pharmacists are often substituted by different pharmacists, making it difficult to build and sustain relationships with patients.

#### 3.3.1. Pharmacists Provide Patient-Care Services That are Impersonal

Participants discussed their experiences with services, such as mail order pharmacy and medication refill reminder phone calls. Participants felt those services limited their interactions with pharmacists and impacted their level of trust.

“Well, it was just the impersonality of it. I mean, I didn’t do anything wrong. I mean, when, and when they would call me to check on things, I mean, they were very caring people on the other end. But it was a voice on the phone, and I don’t care for phones that much. So, you know, I, and they would call, and they would say, well, you know, they would kind of do a little survey. How are you doing?”[PLWH 3].

“And some people don’t like phone calls at all and will never answer the phone, so I feel like we could do better with, yeah, I think about that a lot. We could do better with how pharmacy in our clinic follows patients.”[SW 4].

#### 3.3.2. Pharmacists Provide Insufficient Medication Adherence Support

Part of the services pharmacists provide to people living with HIV is adherence support. Study participants felt that pharmacists can improve in the adherence services they provide patients.

“I see big failures with pharmacies and adherence. So even our mail order pharmacy, which is, I, I think anyone who has HIV who was enrolled in the mail order pharmacy is enrolled in this special program where a pharmacist is supposed to monitor adherence and call the patient every three months, I think, and just check in. Sometimes I hear my co-workers saying, oh, they haven’t refilled in three months. And no one called from the pharmacy.”[SW 1].

“I would say the biggest difference, and if you sum it all up, it would be a proactive approach to patient care. Where this is how I look it, retail is more reactive like they just go in their retail store, most of the times you’re not going to have the pharmacy reaching out monthly calling patients saying, hey, it looks like you’re due for this refill. A lot of times it’s the patient calling the pharmacy and saying, hey, I need this filled.”[PLWH 4].

#### 3.3.3. Pharmacists Have Limited Time and Resources

Although there is an opportunity for pharmacists to contribute towards patient advocacy, they have limited time and expectancy for how much prescriptions they fill. Pharmacists are expected to focus on generating revenue, which is based on prescription filling rather than patient advocacy.

“Now the bottom line is so tight that I think a lot of pharmacists don’t have time to really be advocates because they’re making $1.25 on a prescription, you know, and you could go over 500 scripts a day.”[SW 4].

#### 3.3.4. Patients Do Not Build Trust with Pharmacists Due to Changes in Pharmacy Staff

Participants also noted that staff issues impact building trust in the pharmacy, especially since pharmacies rotate staff. Due to changes in staff, patients find it difficult to build trust with pharmacists that they do not regularly interact with.

“So how do you build trust? And how do you maintain it. So switching someone around doesn’t build trust. It destroys it. No matter how caring the people are, it makes it very difficult.”[PLWH 2].

“I think if you’re getting a phone call from a pharmacist you’ve never met, it is more difficult to have those trickier discussions with than someone maybe you’ve seen.”[SW 2].

### 3.4. Negative Patient Experiences within the Community Pharmacy Space

Participants described experiences within the community pharmacy space that are negative and impact the care of people living with HIV.

#### 3.4.1. Pharmacist Attitude and Behavior towards People Living with HIV

Participants noted that there is a class disparity between pharmacists and the HIV patients they serve. While pharmacists are educated middle class, their patients may not be similar. Pharmacists need to acknowledge those differences and treat their patients accordingly.

“Well, being nonjudgmental first, I mean, just being understanding of where the person is coming, your patient is coming from because they’re very diverse. You know, it’s not like they’re well educated, middle class.”[SW 3].

Patients have sometimes experienced receiving care from pharmacy staff members that had negative attitudes.

“But I had, sometimes they’re not up to par as far as looking at stuff as they should be. And when, at first, especially, when I first started, when I moved into this apartment where I am now, when I got that, when they knew that they were getting ready to fill out my ARV, and all they, you know, were kind of rude on the phone just, well, we don’t carry this.”[PLWH 3].

“And some, you know, sometimes, a lot of the times, the [pharmacy] staff aren’t respectful and courteous and professional. So that’s another reason to avoid a place.”[PLWH 1].

#### 3.4.2. Breach of Patient Privacy in Pharmacies

Participants acknowledged that breach of confidentiality occurs in pharmacy settings.

“And then I also think, we’ve touched on this a little bit, but the way pharmacies are designed, in general, does not offer privacy. And the way pharmacy staff function, largely breaks privacy just across the board in the worst ways.”[SW 3].

“I mean, after sort of ten years taking the new medications, to go in every time and say, okay, now there’s this, and these are the side effects and, you know. And like, okay, your medicines are ready. Here is the Atripla. Here, you know, in a really loud voice that’s kind of like, you know, you never know who’s going to be around. And I would just prefer them to respect my privacy, and for that to be my decision, how open I want to be about it.”[PLWH 3].

Additionally, there are pharmacies that only serve people living with HIV. Patients felt that some people living with HIV may not want to receive pharmacy services in such locations because it reveals their HIV diagnosis.

“And if a place is known for the treatment of people living with HIV, that’s another reason why people would avoid it. They don’t want to be associated with a place that’s exclusively for people living with HIV.”[PLWH 4].

As part of the study, after participants discussed these barriers to care, they also shared some recommendations to pharmacists providing HIV care and included specific training that will benefit pharmacists and the patient population they serve.

#### 3.4.3. Recommendations for Pharmacists Providing HIV Care

Participants discussed activities that pharmacists can be involved in to provide better care for people living with HIV. Pharmacists can become familiar with resources within their community that address patients’ barriers to care, they can protect the privacy of their patients, as well as build relationships with the patients and HIV care providers in the community.

Build rapport with patients to encourage engagement in HIV care

Pharmacists can use their accessibility to patients as an advantage to address barriers to patients’ engagement in HIV care. Pharmacists have patient contact information and can see patients face-to-face when they visit the pharmacy. Participants view pharmacists having a closer relationship with their patients as a positive action that helps their patients.

“I think what my pharmacists were able to do, they knew me well enough to look and say, you know, is everything all right? So they could tell if things weren’t going well, you know, for whatever reason.”[PLWH 3].

“[…] the stigma surrounding it can be very, very isolating for people, if you just have people who like you feel care about you, I think that can make a difference to you staying linked in to care.”[CP 3].

b.Create a safe space in the community pharmacies for patients

In addition to trusting the pharmacists, participants believe that trusting the pharmacy environment is just as important. Community pharmacies are expected to be welcoming, judgment-free, and an understanding environment where patient needs are met.

“And so I think meds are such a huge, huge part of HIV care that, and patients know that, that they are like very concerned that they have a good pharmacist and a good pharmacy and someone that they can ask really important questions to and still feel trusted.”[CP 2].

The participants made recommendations to pharmacists on how to make their patients feel welcomed. Pharmacists can do this by understanding their patient population, and being open-minded, willing to learn, and not being judgmental.

“As long as they stay open to change and open to understanding each person that approaches their counter is a different person, is a different person with a different circumstance.”[PLWH 1].

“So as long as they stay open and not try to always, you know, pre-assume anything by when they first look at you no matter what age you are, no matter what color you are, no matter any of that, as long as they don’t try to pre-assume anything, then they will have a much better day of doing their job.”[PLWH 3].

“And I think knowing that there is immense stigma around HIV is super important for any provider, whether it’s pharmacist or other community providers, to be aware of. I think being very aware of the importance of engagement in care, or retention in care, is really important, because for some pharmacists, they, you know, might be, a lot of patients are maybe, their chronic health conditions are maybe not as potentially problematic as HIV can be for someone’s life and their health.”[SW 4].

c.Protect the privacy of patients with HIV

In the community pharmacy setting, there are many opportunities for the unintended disclosure of a patient’s HIV status. Participants discussed some of these challenges and initiatives that pharmacies can use to protect patient privacy, such as using untraceable phone numbers to call patients, and providing patients with medication bottles that do not disclose the owner or the content of the bottle. Below are some verbatim comments that discuss how some participants feel about receiving phone calls from the pharmacies and having prescription bottles in their house.

“And some of them are like absolutely not and will go ballistic if the pharmacy calls them, because they’re so, so scared about somebody finding out.”[SW 2].

“I just didn’t want the medication bottle in my house. And I used to take, I used to put my pills in another more discreet bottle. So if people had that option, I think that would ease some people.”[PLWH 5].

“It’s nice to have a little area, if you’re waiting, that you can sit down and that’s semi-private. I think, in general, most pharmacists probably are, in terms of, you know, they’re not going to like yell your name out and what you’re there for, you know.”[SW 1].

The participants discussed ways in which pharmacists decrease the risk of disclosing patients’ HIV status by respecting patients’ wishes and providing services in a discreet manner by not having signs that disclose a place as a location where people living with HIV receive their services.

“We don’t have any signage on our like delivery drivers, things like that. So we try to make sure everything is unmarked. We also are very receptive to patient requests.”[CP 2].

“But, yeah, as far as stigma goes, I would say pharmacists just trying to be as discreet as possible when discussing things with patients [...] encouraging other employees to not be gossipy or indiscreet about talking about people living with HIV.”[CP 3].

d.Hire diverse pharmacy staff

Participants discussed that they would feel more comfortable in a pharmacy that has a diverse staff population. They believe that staff that have similar life experiences as them will treat them with a better understanding. The participants specifically discussed a wish to see more black or trans staff.

“But having gay people working there [a community pharmacy] was really helpful [...] in fact, I think if I wouldn’t have felt that, I would have felt more isolated.”[PLWH 2].

“And all the people who work at (Pharmacy) were plugged into that community. There were a lot of gay and lesbians there, who worked there, so it just felt very comfortable to me.”[PLWH3].

“But there’s only one black person working in there. I’m just noticing how the disparities are, you know, they’re not reaching as many people as they possibly can reach, because they’re not hiring a more diverse team.”[PLWH 5].

e.Provide pharmacy staff with information about transgender people

Patients felt that pharmacists needed additional training on how to take care of their transgender patients that are living with HIV. These patients need specialized care and pharmacists need to have more information about how to care for them.

“I would like for them to have a lot of information on trans people.”[PLWH 3].

“But with trans people, you have to walk this fine line of being very respectful and not looking at them as though they were in a circus. And so that would also help in the pharmaceutical industry.”[PLWH 5].

f.Trainings for community pharmacists to consider

Participants felt that pharmacists are experts at providing drug therapy management; however, when it comes to HIV care, pharmacists need additional HIV-specific knowledge about insurance. Additionally, participants felt pharmacists needed cognitive skills to care for transgender individuals as well as how to communicate with patients in a sensitive manner.

“And that’s still going to all take place, but I figure that some of it, you know, if the pharmacists had like an ongoing training, you know, it’s one thing to be able to supply people with their medications but have a little bit more compassion and human interaction other than just explaining to them what this medication is for.”[SW 4].

i.Train pharmacy staff on how to communicate with vulnerable populations

Participants felt that pharmacists can be better at communicating sensitive information to their patients, especially considering the vulnerable population that are infected with HIV, such as people who use drugs and patients who belong to the gay community.

“And I suppose, because, you know, well, HIV is, I mean, not only associated with gay people and intravenous drug use, and it’s all the things that people can’t have attitudes, so even if people don’t have an attitude, I would probably be sensitive to it and watching out for it. So I think that’s really important to, you know, completely just treat it like a medical issue, and maybe some sensitivity training for those people who, you know, aren’t exposed to the gay community.”[SW 2].

“I think pharmacists assume too much. It’s like, oh, do they(patients) even read their prescription bottle? You know, there’s a, and then there can be language barriers and mental health barriers.”[SW 3].

ii.Train pharmacy staff about HIV insurance

Participants of our study perceived pharmacists will benefit from learning about insurance. The expectancy is not for pharmacists to be knowledgeable about everything related to HIV care, but that they should be able to answer questions that often arise about medication cost.

“So sometimes that takes a little bit of teaching the pharmacy that they are connecting with to, you know, educate them about, okay, what does it mean when somebody has coverage through the ADAP program and how, you know, this is how you run into barriers filling it if this, and coming up with like a huge bill, and then there is something wrong with the billing, or trying to prepare a pharmacy that might not usually fill specialty medications as to like keeping those stocked and in expectation of needing to have those available for patients.”[SW 1].

“You need to know simple things like that with the financial, you need to know like the prescription assistance programs, and then you need to know some basics. You are not expected to know, you know what I mean, everything, but those questions that pop up…”[SW 3].

## 4. Discussion

This study showed the perspectives of pharmacists, pharmacy staff, and the community pharmacy environment that act as barriers to patients utilizing pharmacy services for their HIV care. These barriers were related to a perceived lack of confidence in the community pharmacist’s knowledge of HIV care and the limitations to their role in HIV care. Participants identified operational activities, such as staffing changes, that act as barriers to HIV care, as well as time constraints in the pharmacy. Participant perceptions of solutions for improving HIV care in the community pharmacy focused on improving the relationship between pharmacists and patients, ensuring that the community pharmacy is a private and safe space for patients and having a diverse pharmacy staff that is equipped to take care of the diverse and marginalized HIV population, such as transgender individuals. The perceived solutions also addressed barriers to pharmacists’ HIV insurance knowledge and the attitudes of pharmacists towards people living with HIV.

Tsent et al. published a guideline for the role of pharmacists in HIV care, which focused mainly on drug therapy management [18]. Drug therapy is not the only factor that impacts retention in patient care. Additionally, many patients in the community are unaware that pharmacists have been extensively trained to provide basic healthcare services beyond drug therapy management, such as providing blood pressure screenings, educating patients with diabetes on the effective use of glucometers, and ultimately providing an interpretation of these readings and diagnostic tools [42,43,44]. Despite their ability to provide these services, pharmacists feel underutilized within the community setting [20,45,46]. Participants in our study were confused about the role of pharmacists in HIV care; they acknowledged that pharmacists are the drug experts, but in a way that limits their scope of practice. When the participants discussed services that pharmacists can provide, such as referral to community resources, they questioned if it was within the scope of community pharmacy practice. When the participants recommended that pharmacists take an active role in addressing the non-medical needs of patients, they addressed the pharmacists as a patient-advocate social worker. For example, when patients think of pharmacists, all that comes to mind is drugs; however, pharmacists discussed advocacy roles that pharmacists participate in and how pharmacists can be trained in providing additional support to people living with HIV. Overall, there seems to be a disconnect between what pharmacists are trained to do, what they do within the pharmacy setting, what patients think they do, and what patients wished they did more. Although this conundrum may be frustrating, it is an opportunity for pharmacists to increase the visibility of their profession by contributing to public health efforts to improve the health outcomes of patients living with HIV.

Our study of community pharmacists providing HIV care revealed that pharmacists are willing to take more roles in HIV patient care to address health care delivery system challenges. For pharmacists to expand their role in HIV care, they need to be competent in the type of care they are delivering. A literature review of pharmacists’ confidence in providing health promotion services was mixed [42], and notably revealed their lower level of confidence in HIV-related health promotion [42,47]. Patients in our study were not confident that pharmacists were knowledgeable about their patients’ needs, especially in addressing the needs of patients with cognitive delays.

Another patient barrier that participants believed pharmacists are unaware of includes cost barriers associated with HIV and patient insurance options. Cost of medication and HIV services has been a well-studied barrier to retention in HIV care [8,48,49,50]. In one study, people living with HIV specified insurance as one of their most common barriers to care [49]. People with HIV are often eligible for programs, such as the Ryan White Program that covers drug costs [49,51]. The Ryan White program is an AIDS Drug Assistance program (ADAP), the nation’s primary prescription assistance program for low-income uninsured individuals living with HIV [52]. However, patients as well as their health care providers may not be familiar with the financial assistance programs available to patients. The social workers in this study discussed their experiences with pharmacists that are not knowledgeable about insurance programs, including ADAP. Our study’s pharmacists were also not confident that pharmacists that do not routinely fill HIV prescriptions will be familiar with drug coverage options for people living HIV.

Pharmacists need to become familiar with the insurance programs available for people living with HIV and provide them with information on how to apply. Pharmacists can also have information about drugs that have a lower cost than currently prescribed medications. In one study, a pharmacist contacted a patient’s case manager about incomplete insurance paperwork to ensure payment of antiretroviral medication costs [20]. Pharmacists can work collaboratively with social workers to learn more about their patients’ insurance options to provide optimal HIV care.

For pharmacists to be able to connect with patients similarly to social workers, they need to develop close relationships with their patients. Patients that have a better relationship with their healthcare service providers are more likely to be retained in HIV care [8,53]. Previous studies show that social workers have very close bonds with their patients [53]. By building trusting relationships with patients, pharmacists are able to assess the immediate patient needs and provide them with either emotional support or information about resources where their needs can be met.

Although participants appreciate the accessibility of community pharmacies, they had negative experiences with pharmacy staff. Participants believe that pharmacy staff can often be judgmental, lack empathy, and interact with patients in ways that shows their limited experiences with HIV care. These characteristics negatively impact the relationships pharmacists have with their patients. Consequently, if patients cannot trust their pharmacists, it affects their overall health outcome [54,55]. People living with HIV already experience stigma in their communities; therefore, there is a stronger need in this population for the pharmacy setting to be a safe space.

A study of pharmacy students showed that they still had negative general attitudes toward HIV, perceived themselves as having greater negative attitudes on specific aspects of interacting with persons with HIV, and possessed significantly greater willingness to provide services to patients with asthma than those with HIV/AIDS [56]. There is a possibility that pharmacists that are practicing also carry similar negative attitudes towards people living with HIV. Participants in this study recommended that pharmacies have staff that share a similar experience to people living with HIV; this will make the staff have more empathy. The study also recommended that pharmacists should make it known to their patients that their HIV status will not be disclosed beyond what is necessary for their care. The pharmacists and their staff need to become aware of the real-life consequences that the unintended disclosure of a patient’s HIV status can have on the patients and train their staff to keep all information confidential even when discussing patients in the pharmacy. Future studies can examine the stigma pharmacists have towards people living with HIV and design interventions that will educate pharmacists about the impact of their stigma and how to address their negative feelings towards these patients.

A proposed solution for community pharmacies to build trust with their HIV patient population is to hire staff that are more diverse and thus experienced in understanding the challenges patients face when living with HIV. These shared experiences are a good catalyst for building relationships between patients and their health caregivers. Pharmacy schools have emphasized admitting more diverse student bodies [57] however, these efforts are not seen in the pharmacy setting, as most pharmacies are still operated by non-Hispanic white pharmacists. Staff diversity should be advocated for as a target of community pharmacies. HIV disproportionately affects minoritized groups, such as men who have sex with men [58] as well as black and Latino men [59]. For pharmacists to be able to connect with patients similarly to social workers, they need to develop close relationships with PLWH.

### Study Limitations

The strength of this study is that it explored the concerns with HIV care through the perspective of different stakeholders. This study also used a conceptual framework to guide the data collection and data analysis, ensuring that all study activities were relevant to the goal of the study. Nonetheless, the findings of the study should be viewed in light of several study limitations that should be considered when assessing the results and significance of the study. The study setting and sample is from a midwestern state in the United States, which has a limited diversity pool of individuals living with HIV, community pharmacists that care for people living with HIV, and social workers that are trained in linking and retaining patients in HIV care. Only two out of 15 of the study participants were people of color, and only one of the five people living with HIV identified as men-who-have-sex-with-men, despite these two groups being the most disproportionately infected individuals living with the virus. Marginalized populations need to be represented in the data. However, since this study did not target a specific racial group, gender, or sexual orientation, individuals that met the inclusion criteria were invited to participate. Additionally, this study will benefit from a larger quantitative study that examines these barriers to care in community pharmacies across the United States.

## 5. Conclusions

There is a potential for community pharmacists in the United States to be more involved in the various stages of HIV care, especially in helping patients to be retained in HIV care. To date, however, there has been no literature referring specifically to pharmacists’ role in the community addressing barriers that impact all aspects of the patient’s care, including their medication management. Community pharmacies provide a range of public health services, such as immunizations, counseling for diseases, medication therapy management, and testing for chronic conditions [60]. In HIV care, the pharmacists’ services are limited to medication adherence [61]. This study has shown that there are many barriers that patients face that inhibit them from accessing care or engaging with their HIV care. The barriers focused on the perceived role of pharmacists in HIV care that are limited to their drug expertise, despite their ability to advocate for patients. Patients also had experiences in the pharmacy that affected their trust and confidence in pharmacists’ knowledge of their needs. The participants of this study recommended that pharmacy staff receive training on how to communicate and take care of marginalized populations. There is also a need for the community pharmacy setting to be a safe space for people living with HIV, where they are not worried about their privacy and feel that they are not being judged by pharmacy staff. The study participants recognized the limited resources and time constraints that pharmacists operate under as additional barriers to HIV care. Pharmacists have the training, desire, and accessibility to patients and can work with individuals providing HIV care in their communities, social workers, to address barriers to care. The pharmacist and community pharmacy can become better equipped to provide HIV care in a way that encourages patients to link to care and remain retained in HIV care.

## Figures and Tables

**Table 1 pharmacy-09-00178-t001:** Eight sub-themes each accompanied by a representative quote. Quotes were selected to represent the variety of issues discussed under that subtheme. Participant subgroup denoted in parentheses.

**a. Perceptions of the role of community pharmacists in HIV care**	
Pharmacists have a limited role in HIV care as drug experts	“We [social workers] are kind of like the thread that weaves through the patient’s life and their medication and their primary care and their specialty care. And then the pharmacist has kind of like this more narrow picture and also can’t do anything about it, and that’s where I think social work comes in for us now.” [SW 1]
Pharmacists act as patient advocates	“So I would see an injustice that I felt like people weren’t getting proper care, or they weren’t, you know, getting help from their insurance or whatever, so then I would go. So I became a little bit of a social worker myself.” [CP 1]
**b. Perceptions of Pharmacists’ Knowledge about HIV resources**	
“Well, I don’t know in general…how much of current research that pharmacists are equipped with…Like, again, went to a chain pharmacy, I don’t know if I would just have that automatic confidence.” [PLWH 2] “They (pharmacists) have no idea about ADAP (AIDS Drug Assistance Program). They have no idea about any kind of prescription assistance.” [SW 2]	
**c. Perception of pharmacy operation and services**	
Pharmacists provide patient-care services that are impersonal	“I had to do the mail order thing for a while because of my insurance. I hated it.” [PLWH 5]
Pharmacists provide insufficient medication adherence support	“I think most of my experience with community pharmacists has been that type of like coordination and not so much the adherence support and stuff.” [SW 3]
Pharmacists have limited time and resources	“You know what I mean? Like can they (Pharmacists) facilitate care? Do they have like here it just seems like our pharmacies are so like backed up and understaffed.” [SW1]
Patients don’t build trust with pharmacists due to changes in pharmacy staff	“And if you don’t trust the pharmacist, you’re just going to be like, yeah, I’m taking my med every day. I don’t really want to talk to you.” [SW 5]
**d. Negative experiences within the community pharmacy space**	
Pharmacist attitude and behavior towards people living with HIV	“Probably when I got to know the pharmacist, it would be okay, but I would, perhaps, how to say this, be not 100% confidence about their attitude about people with HIV.” [PLWH 1]
Breach of privacy in pharmacies	“We’ve had people transfer their meds into this pharmacy because they live in very small rural communities, and even though HIPAA is a thing, sometimes in small communities things can still get out.” [CP 4]

## Data Availability

Data is available from corresponding author on request.

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
