# Peer review of "Patients, Social Workers, and Pharmacists’ Perceptions of Barriers to Providing HIV Care in Community Pharmacies in the United States"

_pharmacy, 2021, doi:10.3390/pharmacy9040178_

Round 1

Reviewer 1 Report

While this is an important topic, I spent the first half wondering how a community pharmacist was defined and what country we were talking about. It was much later clear that this was in the US. I'm also not sure if the background section is relevant to only the US or worldwide. In addition, there was reference to this being part of a larger study, but that study was never defined nor how this work fit into that. This sample size was extremely small and I wasn't sure if this was all in one state or across the US. I'm not sure how saturation was reached with 5 in each group if we are looking at the US as a whole, let alone mixing urban and rural areas. Maybe this small sample size is sufficient if it is complementing other work. While the findings were interesting, I was left with a lot of questions in the end and can't say there were credible on their own. It would have to be supplemented with the larger study, which I'm assuming is quantitative or it the context and sampling has to be better defined to justify this work.

Author Response

Reviewer 1

While this is an important topic, I spent the first half wondering how a community pharmacist was defined and what country we were talking about.

Thank you, for your comment. We have defined a community pharmacist as, “a pharmacist practicing in a registered premise that is approved and licensed to dispense medication, provide professional counseling, and perform pharmacy services to people in a local area. This is information is included on page 2.

It was much later clear that this was in the US. I'm also not sure if the background section is relevant to only the US or worldwide.

We are talking about the United States. We have included that in the abstract; introduction (page 1); methods (page 3), discussion (page 11), study limitations (page 13), and conclusion (page 14) to specify that the study is based on community pharmacies in the United States.  

 In addition, there was reference to this being part of a larger study, but that study was never defined nor how this work fit into that.

Thank you for the comment. The overall study aims to identify these barriers and explore ways in which pharmacists and social workers can collaborate interprofessional to address these barriers. This paper focuses on the barriers to care experienced specifically in the community pharmacy and the recommendations on how community pharmacists can address those barriers. We have included this information on page 2.

 This sample size was extremely small, and I wasn't sure if this was all in one state or across the US.

This study was conducted in one Midwestern State. This information was included in the study limitation on page 13. But we included this information earlier in the methods section on page 2.  

 I'm not sure how saturation was reached with 5 in each group if we are looking at the US as a whole, let alone mixing urban and rural areas.

Thank you for your comment. We have included in the methods section that this sample is from a Midwestern State on page 2. We included information about data saturation on page 4. The research question focused specifically on linkage and retention in care with an even narrower focus on pharmacy, as such saturation was achieved with small sample size. For example, while interviewing the third pharmacist we started to recognize that no new themes were emerging. Similarly, patients when asked about their experiences in the community pharmacy they raised similar concerns about a breach of privacy, being judged by staff, and having insurance issues. Therefore, due to the study design and research question, with the small sample size, we were able to reach saturation.

Maybe this small sample size is sufficient if it is complementing other work.

Thank you for your comment. This is a qualitative work as such the sample size selection is based on when saturation is achieved. With the current sample, no new themes were emerging as such there was no need for further interviews of more stakeholders. Data analysis research questions were very focused (role of pharmacy in retention to care), and therefore saturation could be achieved with smaller sample size. Although this is not complementing other work, the directed content analysis process ensures there is a direction and guidance to the structure of the study.

While the findings were interesting, I was left with a lot of questions in the end and can't say there were credible on their own. It would have to be supplemented with the larger study, which I'm assuming is quantitative or it the context and sampling has to be better defined to justify this work.

We have included your concerns in our conclusion as a need to address the findings with a larger study that has a quantitative approach. We included the sentence “this study will benefit from a larger quantitative study that examines these barriers to care in community pharmacies across the United States.” Exploratory work is needed first before tackling a larger quantitative study. There is a need to first understand the barriers from stakeholder perspectives to provide researchers with the directions to move forward with the work. We included this information on page 13.

Reviewer 2 Report

Patients, Social workers, and Pharmacists’ Perceptions of Barriers to Providing HIV Care in Community Pharmacies

General Comments

This is an important piece of scholarly work. However, it requires some work to make it stronger. There were several unsupported claims and generalizations.  Please see below specific comments:

Abstract

  • The abstract is concise and clear. However, one wonders why only transgender people were mentioned instead of all sexual and gender minorities.
  • The authors should contextualize the title because these perceptions cannot be generalized.

Introduction

  • The introduction looks good and engages with relevant literature. However, the authors should endeavor to contextualize the literature rather than generalizing. For example, “a study in the US or Kenya found that…”. See examples from your manuscript: “Currently, more people living with HIV are receiving care in the community setting rather than the hospital setting [21]. Conveniently, approximately 70% of people in rural areas live within 15 miles of a pharmacy, and 90% of people in urban areas live within two miles of a pharmacy [22, 25].”
  • These two sentences can be summarized as one: “The care people with HIV receive is examined on a continuum where a person moves from not knowing their HIV status to becoming virally suppressed [5-7]. Retention in HIV care occurs when 34 patients utilize HIV care services to move along the HIV continuum of care [8, 9]”.
  • These statements should be rephrased: “When people with HIV are retained in care, more than 90% receive antiretroviral therapy and 36 more than 80% achieve complete viral suppression [10].
  • I would say ‘substance use disorder’. Substance use in line 40 should be put in a context.
  • I would suggest the authors merge 1.3 and 1.4.
  • The following statement is vague. Contextualize your study: “To our knowledge, this is the first study to examine barriers to retention to HIV care in the community pharmacy setting and to utilize three stakeholder groups: patients, pharmacists, and social workers.” In the US or globally?

Methods

  • I would suggest you delete the materials on the heading.
  • Study design and sample: the authors should endeavor to go straight to the point here. Tease out the study design and sample clearly. Please summarize this section and tease out the logical flow. Please describe clearly how you used the PCMH model in your study. Did this model inform the data analysis? There is no justification for the small sample size.
  • Data collection: the authors should describe the consent process and interviewing language. I would suggest that the authors should focus on the interviewing process in the first paragraph although this was seen in the third paragraph.

Data Analysis

  • What are PCMH constructs? Please describe how PCMH model1 informed your data analysis?
  • The following sentence should have come earlier: “NVIVO Version 12 (QSR International-Melbourne) was used to catalog coded transcripts”. I believe the authors used NVIVO to code and manage the transcripts. This should be stated clearly. The data analysis description is very thin. The following is the only sentence about data analysis: “Directed content analysis [41] was used to analyze the data deductively”. Please explain this analysis process to your readers. No information about how the notes that the interviewer took was used during the data analysis process. Did you present participants’ views subjectively? How was PCMH model used during data analysis? It seems to me that interpretivist approach was adopted during data analysis. No information about the themes and it seems that thematic analysis was performed.

Results

  • I think the authors can incorporate the sub-themes (including quotes) in the tale to the result section.
  • Please rephrase this sentence: “Participants in the study including pharmacists say patient advocacy as something outside of their limited scope.”
  • Please edit some participants’ quotes for logical flow.
  • Most of the sub-themes are short. The authors need to unpack some of them. Your readers need to understand your position as well not only what you were told.
  • The sub-themes are too much. Some of them can be merged to make meaning sense.
  • I also did not see how PCMH model was adopted during analysis to inform the study findings.
  • The sections need to be reworked to make it stronger.

Discussion

  • Please contextualize your claims: “To our knowledge, this is the first study that examines pharmacy-specific barriers to HIV care within the community pharmacy that may impact patient retention in HIV care”. In Kenya, US, or South Africa? Also, this has been mentioned before in the introduction. I would delve into the findings’ discussion immediately.
  • I would suggest that the authors discuss each key findings in one paragraph rather than the multiple sub-headings in the section.
  • How the authors used the conceptual framework to guide data analysis and results is not clear. Kindly clarify.
  • This section needs some work to make it stronger.

Study limitations

  • Please write the actual numbers instead of 27% people of color as well as 20% MSM.

Conclusions

  • The conclusion should include the summary of the key findings. There is a lot of generalizations that need to be contextualized.

Author Response

Reviewer 2

General Comments

This is an important piece of scholarly work. However, it requires some work to make it stronger. There were several unsupported claims and generalizations.  Please see below specific comments:

Thank you for your feedback about our study.

Abstract

  • The abstract is concise and clear. However, one wonders why only transgender people were mentioned instead of all sexual and gender minorities.

Thank you for your comment. The study participants specifically mentioned individuals that identify as transgender and thus were referenced in the abstract.

  • The authors should contextualize the title because these perceptions cannot be generalized.

This is a qualitative study. As such, the goal is not to generalize our findings. We have included in the heading that this study is in the United States. As such the title is “Patients, Social workers, and Pharmacists’ Perceptions of Barriers to Providing HIV Care in Community Pharmacies in the United States”

Introduction

  • The introduction looks good and engages with relevant literature. However, the authors should endeavor to contextualize the literature rather than generalizing. For example, “a study in the US or Kenya found that…”. See examples from your manuscript: “Currently, more people living with HIV are receiving care in the community setting rather than the hospital setting [21]. Conveniently, approximately 70% of people in rural areas live within 15 miles of a pharmacy, and 90% of people in urban areas live within two miles of a pharmacy [22, 25].”

Thank you for your comment, we have specified that these studies are from the United States and included the clarification in our text. We have changed how we reframed the literature we included.

  • These two sentences can be summarized as one: “The care people with HIV receive is examined on a continuum where a person moves from not knowing their HIV status to becoming virally suppressed [5-7]. Retention in HIV care occurs when 34 patients utilize HIV care services to move along the HIV continuum of care [8, 9]”.

Thank you for your feedback, we have combined the sentence. It is now rewritten asPeople living with HIV move along a continuum of care [5-7] from not knowing their status to becoming retained in HIV and then virally suppressed [8, 9].”

  • These statements should be rephrased: “When people with HIV are retained in care, more than 90% receive antiretroviral therapy and 36 more than 80% achieve complete viral suppression [10].

Thank you for your feedback, we have rephrased the sentence to say that more than 80% of the individuals who are retained in HIV care achieved viral suppression.

  • I would say ‘substance use disorder’. Substance use in line 40 should be put in a context.

Thank you, we have made the change.

  • I would suggest the authors merge 1.3 and 1.4.

Thank you, we have merged sections 1.3 and 1.4.

  • The following statement is vague. Contextualize your study: “To our knowledge, this is the first study to examine barriers to retention to HIV care in the community pharmacy setting and to utilize three stakeholder groups: patients, pharmacists, and social workers.” In the US or globally?

Thank you for your comment, we have clarified that this is in the United States alone.

Methods

  • I would suggest you delete the materials on the heading.

Thank you, we have made the deletion. 

  • Study design and sample: the authors should endeavor to go straight to the point here. Tease out the study design and sample clearly. Please summarize this section and tease out the logical flow.

Thank you for your comment, we have revised the study design and sample section to be more succinct.

Please describe clearly how you used the PCMH model in your study. Did this model inform the data analysis?

The PCMH model informed interview question guide and the data analysis. We described in the

There is no justification for the small sample size.

Thank you for your comment. This is a qualitative work as such the sample size selection is based on when saturation is achieved. We included information about data saturation on page 4. With the current sample, no new themes were emerging as such there was no need for further interviews of more stakeholders. Data analysis research questions were very focused (role of pharmacy in retention to care), and therefore saturation could be achieved with smaller sample size.

  • Data collection: the authors should describe the consent process and interviewing language.

Thank you, we included the consent information under the 2.3.1. Ethics and compliance statement section.

 I would suggest that the authors should focus on the interviewing process in the first paragraph although this was seen in the third paragraph.

Thank you, we have re-organized the section.

Data Analysis

What are PCMH constructs?

Thank you for your comment. Under the data collection section on page 3, we described the PCMH constructs as follows:

“The semi-structured interview was designed using adapted constructs of the PCMH model – experiences, activities, interventions, and outcomes – that have an impact on the utilization of HIV care services in the community pharmacy setting. This paper focuses on the experiences and activities constructs to examine barriers to care in the community pharmacy setting. Questions explored patient experiences in seeking care in the community pharmacy setting and the barriers experienced within the environment. Pharmacists shared their experiences providing HIV care in the community pharmacy setting. Social workers also shared their experiences in working with community pharmacists to provide care to people living with HIV and the barriers their patients experience when receiving care within the community pharmacy environment. Activities pharmacists can provide in the pharmacy setting to improve utilization of HIV care from the purview of the study stakeholders were explored in-depth. In this study, we defined barriers to retention in care as any factor identified by study participants that hinder patients to continuously utilize pharmacy services and continuously seek HIV care.”

Please describe how PCMH model1 informed your data analysis?

Thank you for the question, on page 4 we describe how the model informed data analysis as follows:

“Directed content analysis [41] was used to analyze the data deductively. Directed content analysis starts with a model, existing theory, or research findings to guide data collection and analysis [41]. The researchers followed the following steps in their directed content analysis: first, reading the transcripts several times to achieve immersion; then reading the data line by line to capture ideas, and coding and organizing the themes and categories per the mapping of the constructs guided by PCMH model. Factors that acted as barriers to retention in care were coded into four constructs of the PCMH: (a) interventions (b) activities, (c) experiences, and (d) outcomes. Specific themes emerging under each construct emerged from the data. The identified themes within each construct were explained with direct verbatim quotes to further clarify the themes. The researchers allowed for the development of themes inductively expand their exploration of the data. The findings presented in this paper focus on the experiences and activities constructs as well as the inductive findings.”

The following sentence should have come earlier: “NVIVO Version 12 (QSR International-Melbourne) was used to catalog coded transcripts”. I believe the authors used NVIVO to code and manage the transcripts. This should be stated clearly.

Thank you, we have moved the sentence earlier and clarified the sentence.

The data analysis description is very thin. The following is the only sentence about data analysis: “Directed content analysis [41] was used to analyze the data deductively”. Please explain this analysis process to your readers.

Thank you, we have explained directed content analysis to our readers in the data collection and data analysis section on pages 3 and 4.

 No information about how the notes that the interviewer took was used during the data analysis process.

Thank you we have included this additional information about the interview notes

 “The notes taken during the interviews were used during the data analysis process to provide additional context to the transcripts and participants’ quotes” on page 4.

Did you present participants’ views subjectively? How was PCMH model used during data analysis? It seems to me that interpretivist approach was adopted during data analysis. No information about the themes and it seems that thematic analysis was performed.

Thank you for your comment. on page 4 we describe how the model informed data analysis as follows: “Directed content analysis [41] was used to analyze the data deductively. Directed content analysis starts with a model, existing theory, or research findings to guide data collection and analysis [41]. The researchers followed the following steps in their directed content analysis: first, reading the transcripts several times to achieve immersion; then reading the data line by line to capture ideas, and coding and organizing the themes and categories per the mapping of the constructs guided by PCMH model. Factors that acted as barriers to retention in care were coded into four constructs of the PCMH: (a) interventions (b) activities, (c) experiences, and (d) outcomes. Specific themes emerging under each construct emerged from the data. The identified themes within each construct were explained with direct verbatim quotes to further clarify the themes. The researchers allowed for the development of themes inductively expand their exploration of the data. The findings presented in this paper focus on the experiences and activities constructs as well as the inductive findings.”

Results

  • I think the authors can incorporate the sub-themes (including quotes) in the tale to the result section.

Thank you, the results are presented in this order: themes, description of themes, subthemes, description of subthemes, and quotations. This is an easy flow for readers to get a full picture of the different study findings.

  • Please rephrase this sentence: “Participants in the study including pharmacists say patient advocacy as something outside of their limited scope.”

Thank you, we have rephrased the sentence.

  • Please edit some participants’ quotes for logical flow.

Thank you, we have made edits to the quotes.

  • Most of the sub-themes are short. The authors need to unpack some of them. Your readers need to understand your position as well not only what you were told.

Thank you for the comment, we

  • The sub-themes are too much. Some of them can be merged to make meaning sense.

Thank you for the comment. We merged some of the themes and subthemes to make them more cohesive. We have a total of four themes with an average of two subthemes each. We have decreased some of the quotes to make

I also did not see how the PCMH model was adopted during analysis to inform the study findings.

Thank you for your comment. on page 4 we describe how the model informed data analysis as follows: “Directed content analysis [41] was used to analyze the data deductively. Directed content analysis starts with a model, existing theory, or research findings to guide data collection and analysis [41]. The researchers followed the following steps in their directed content analysis: first, reading the transcripts several times to achieve immersion; then reading the data line by line to capture ideas, and coding and organizing the themes and categories per the mapping of the constructs guided by PCMH model. Factors that acted as barriers to retention in care were coded into four constructs of the PCMH: (a) interventions (b) activities, (c) experiences, and (d) outcomes. Specific themes emerging under each construct emerged from the data. The identified themes within each construct were explained with direct verbatim quotes to further clarify the themes. The researchers allowed for the development of themes inductively expand their exploration of the data. The findings presented in this paper focus on the experiences and activities constructs as well as the inductive findings.”

The sections need to be reworked to make it stronger.

Thank you for the comment we have reworked the section.

Discussion

Please contextualize your claims: “To our knowledge, this is the first study that examines pharmacy-specific barriers to HIV care within the community pharmacy that may impact patient retention in HIV care”. In Kenya, US, or South Africa? Also, this has been mentioned before in the introduction.

Thank you, we have removed this sentence.

I would delve into the findings’ discussion immediately.

Thank you for your comment, we have removed the introductory sentences and delved immediately into the findings.  

I would suggest that the authors discuss each key findings in one paragraph rather than the multiple sub-headings in the section.

Thank you, we have revised the discussion section to discuss key findings in one paragraph rather than using multiple sub-headings in the sections.

How the authors used the conceptual framework to guide data analysis and results is not clear. Kindly clarify.

Thank you for your comment. on page 4 we describe how the model informed data analysis as follows: “Directed content analysis [41] was used to analyze the data deductively. Directed content analysis starts with a model, existing theory, or research findings to guide data collection and analysis [41]. The researchers followed the following steps in their directed content analysis: first, reading the transcripts several times to achieve immersion; then reading the data line by line to capture ideas, and coding and organizing the themes and categories per the mapping of the constructs guided by PCMH model. Factors that acted as barriers to retention in care were coded into four constructs of the PCMH: (a) interventions (b) activities, (c) experiences, and (d) outcomes. Specific themes emerging under each construct emerged from the data. The identified themes within each construct were explained with direct verbatim quotes to further clarify the themes. The researchers allowed for the development of themes inductively expand their exploration of the data. The findings presented in this paper focus on the experiences and activities constructs as well as the inductive findings.”

This section needs some work to make it stronger.

Thank you for your comment we have worked on the section to strengthen it.

Study limitations

  • Please write the actual numbers instead of 27% people of color as well as 20% MSM.

Thank you, we have changed the presentation from percentages to numbers.

Conclusions

The conclusion should include the summary of the key findings.

Thank you for the comment. We included a summary of the key findings in the discussion.

There is a lot of generalizations that need to be contextualized.

Thank you for your comment. We revised the generalized statements and contextualized the conclusion to refer to the study aim and findings.

Round 2

Reviewer 1 Report

Given that the limitations of this study pretty much impact the over study, they should not be stated only in the limitations section. It frames the overall study, especially that this sample is not representative of the population that is HIV positive in the US. Those populations might have very different experiences. In addition, how representative is this midwestern state? And are we looking at urban and rural areas in this midwestern state? I'm not satisfied with the responses to how they achieved saturation so quickly, especially if they did not attempt to reach outside of a small setting.

Author Response

Given that the limitations of this study pretty much impact the over study, they should not be stated only in the limitations section.

Thank you for your comment. We included the information that clarifies that the study setting is in a midwestern state in the abstract (page 1), methods (page 3), and the limitations (page 13).

 It frames the overall study, especially that this sample is not representative of the population that is HIV positive in the US. Those populations might have very different experiences.

Thank you for the comment. We agree that people living with HIV have very different experiences. Our study is qualitative, as such the goal is not to generalize to the population of people living with HIV in the United States.  Qualitative research is meant to study a specific issue in a certain group and is focused locally in a particular context, therefore generalizability is usually not an expected outcome of qualitative studies1.  People living with HIV have different lived experiences that cannot be generalized. As qualitative researchers, we acknowledge this limitation in our study and emphasize that the goal is not to collect a general idea of their experiences. Our study uniquely examines their experiences in the context of the community pharmacy setting and captures a setting we are interested in examining. We used a semi-structured interview to guide our findings. We also utilized member checking2,3 and validated our data analysis by returning our findings to our informants for their feedback on how accurately we captured their shared-experiences4.

In addition, how representative is this midwestern state?

Thank you for your question. As defined by the U.S. Census Bureau, the Midwest region includes Iowa, Illinois, Indiana, Kansas, Michigan, Missouri, Minnesota, Nebraska, North Dakota, Ohio, South Dakota, and Wisconsin. Midwestern states have similar HIV infection profiles with lower HIV burden compared to southern states. The midwestern state sampled in our study has a similar HIV prevalence rate as other midwestern states. The range of people living with HIV for the midwestern states in 2018 is between 100 to 300 people living with HIV for every 100,000 persons compared to other states which have over 500 individuals to 100,000 persons living with HIV. A majority of the midwestern states have approximately 100-150 per 100,000 persons and our sampled state has approximately 130 people per 100,000 individuals.

And are we looking at urban and rural areas in this midwestern state?

Our study looked at both urban and rural areas within the midwestern state.

 I'm not satisfied with the responses to how they achieved saturation so quickly, especially if they did not attempt to reach outside of a small setting.

Thank you for your concern. We included information about data saturation on page 4. Our study aim, participant selection, interview guide, and data analysis approach made it feasible to achieve saturation with one-hour semi-structured interviews of 15 participants in our low-HIV burden midwestern state. Sample sizes and settings in qualitative studies are informed by the research question, methods, and/or data analysis approach5-7. A qualitative study can have a sample size as small as one informant 8; a systemic review of HIV qualitative studies showed sample sizes ranging from 3 to 78 participants9. Another study that examined sample sizes in HIV studies found that sample sizes ranged from 1–264 participants, with a median sample size of 14.5 10. Our respondents were sampled from a midwestern state that has a population of 6 million individuals, which is the median population for the eleven midwestern states.

This qualitative study utilized a directed content analysis approach11. A critical aspect of directed content analysis is that the study is more structured and directed by pre-existing theory or frameworks to explore the study aims11. Our research question focused specifically on barriers to retention in care with an even narrower focus on community pharmacy, as such saturation was achieved within the small setting.

Barriers to retention in the pharmacy setting have not been examined before, therefore our study took a guided approach to explore this aim. We made careful selections of informants who are experienced with the study topic in the setting we are exploring. Our study did not cover the whole range of phenomena but presented selected patterns relevant to the study aim. Therefore, due to the study design and research question, with the small sample size within the midwestern state, we were able to achieve saturation across our 15 informants.

References

  1. Leung L. Validity, reliability, and generalizability in qualitative research. Journal of family medicine and primary care. Jul-Sep 2015;4(3):324-327. doi:10.4103/2249-4863.161306
  2. Speziale HS, Streubert HJ, Carpenter DR. Qualitative research in nursing: Advancing the humanistic imperative. Lippincott Williams & Wilkins; 2011.
  3. Creswell JW, Miller DL. Determining Validity in Qualitative Inquiry. Theory Into Practice. 2000/08/01 2000;39(3):124-130. doi:10.1207/s15430421tip3903_2
  4. Birt L, Scott S, Cavers D, Campbell C, Walter F. Member Checking: A Tool to Enhance Trustworthiness or Merely a Nod to Validation? Qual Health Res. Nov 2016;26(13):1802-1811. doi:10.1177/1049732316654870
  5. Malterud K, Siersma VD, Guassora AD. Sample Size in Qualitative Interview Studies: Guided by Information Power. Qualitative Health Research. 2016/11/01 2015;26(13):1753-1760. doi:10.1177/1049732315617444
  6. Malterud K. The art and science of clinical knowledge: evidence beyond measures and numbers. Lancet. Aug 4 2001;358(9279):397-400. doi:10.1016/s0140-6736(01)05548-9
  7. Vasileiou K, Barnett J, Thorpe S, Young T. Characterising and justifying sample size sufficiency in interview-based studies: systematic analysis of qualitative health research over a 15-year period. BMC Medical Research Methodology. 2018/11/21 2018;18(1):148. doi:10.1186/s12874-018-0594-7
  8. Boddy CR. Sample size for qualitative research. Qualitative Market Research: An International Journal. 2016;19(4):426-432. doi:10.1108/QMR-06-2016-0053
  9. Arias-Colmenero T, Pérez-Morente MÁ, Ramos-Morcillo AJ, Capilla-Díaz C, Ruzafa-Martínez M, Hueso-Montoro C. Experiences and Attitudes of People with HIV/AIDS: A Systematic Review of Qualitative Studies. International Journal of Environmental Research and Public Health. 2020;17(2)doi:10.3390/ijerph17020639
  10. Barroso J, Sandelowski M. Sample Reporting in Qualitative Studies of Women with HIV Infection. Field Methods. 2003/11/01 2003;15(4):386-404. doi:10.1177/1525822X03257392
  11. Hsieh H-F, Shannon SE. Three Approaches to Qualitative Content Analysis. Qualitative Health Research. 2005/11/01 2005;15(9):1277-1288. doi:10.1177/1049732305276687